# Digging Through the Complexities of Immunological Approaches in Emerging Osteosarcoma Therapeutics: A Comprehensive Narrative Review with Updated Clinical Trials

**DOI:** 10.3390/biomedicines13030664

**Published:** 2025-03-08

**Authors:** Consolato M. Sergi, Mervin Burnett, Eugeniu Jantuan, Mariam Hakoum, Shawn T. Beug, Roger Leng, Fan Shen

**Affiliations:** 1Division of Anatomic Pathology, Department of Laboratory Medicine, Children’s Hospital of Eastern Ontario, University of Ottawa, Ottawa, ON K1H 8L1, Canada; 2Department of Laboratory Medicine, Stollery Children’s Hospital, University of Alberta, Edmonton, AB T6G 2R3, Canadajant.eugen@yahoo.com (E.J.); fanshen2020@gmail.com (F.S.); 3CHEO Research Institute, University of Ottawa, Ottawa, ON K1N 6N5, Canada; mhako096@uottawa.ca (M.H.);

**Keywords:** immunology, bone, therapy, Osteosarcoma, Omics

## Abstract

Osteosarcoma (OS) is the predominant mesenchymal primary malignant bone tumor in oncology and pathology, impacting a wide age range from adolescents to older adults. It frequently advances to lung metastasis, ultimately resulting in the mortality of OS patients. The precise pathological pathways responsible for OS progression and dissemination are not fully understood due to its heterogeneity. The integration of surgery with neoadjuvant and postoperative chemotherapy has significantly increased the 5-year survival rate to more than 70% for patients with localized OS tumors. However, about 30% of patients experience local recurrence and/or metastasis. Hence, there is a requirement for innovative therapeutic approaches to address the limitations of traditional treatments. Immunotherapy has garnered increasing attention as a promising avenue for tumors resistant to standard therapies, including OS, despite the underlying mechanisms of disease progression and dissemination remaining not well elucidated. Immunotherapy may not have been suitable for use in patients with OS because of the tumor’s immunosuppressive microenvironment and limited immunogenicity. Nevertheless, there are immune-based treatments now being developed for clinical use, such as bispecific antibodies, chimeric antigen receptor T cells, and immune checkpoint inhibitors. Also, additional immunotherapy techniques including cytokines, vaccines, and modified-Natural Killer (NK) cells/macrophages are in the early phases of research but will certainly be popular subjects in the nearest future. Our goal in writing this review was to spark new lines of inquiry into OS immunotherapy by summarizing the findings from both preclinical and current clinical studies examining different approaches.

## 1. Introduction

Osteosarcoma (OS) is a malignant tumor arising from undifferentiated mesenchymal cells and is typically associated with a poor prognosis. Predominantly seen in the metaphysis of long bones, OS affects approximately 4.6 individuals per million worldwide [1,2,3,4,5]. Despite the recent improvements in the 5-year survival rate of patients with localized neoplastic disease using advanced surgical techniques, conventional chemotherapies remain ineffective and toxic for patients with recurrent or metastatic disease [6]. Therefore, there is an urgent need for new and improved treatment.

Understanding the tumor microenvironment (TME) plays a pivotal role in developing and advancing targeted therapies for OS [7]. The unique characteristics of the bone TME, such as its porous mineralized extracellular matrix (ECM) structure and ample availability of nutrients, create an ideal environment for tumor growth. The presence of several types of immune cells such as macrophages, Natural Killer (NK) cells, and T lymphocytes within the TME regulate tumor progression and response to immunotherapies. Furthermore, elucidating the mechanisms of cell migration and adhesion, including the role of adherens junctions and ECM interactions, provides insights into tumor behaviors as well as potential therapeutic targets (Figure 1).

Immunotherapy has emerged as a promising avenue in OS treatment, offering the potential for enhanced efficacy and reduced side effects. Strategies such as immunomodulators, genetically engineered immune cells, and immune checkpoint inhibitors have shown promise in addressing the current challenges in OS treatment. These notable advancements are due to the effective prevention of drug resistance, the reduction of immune evasion, and limiting the off-target side effects through more targeted approaches [8,9,10,11]. In this review, we examine the current landscape of immunotherapy in OS and the importance of understanding their molecular and immunological underpinnings. Through this exploration, we aim to contribute to the ongoing efforts to improve the prognosis and stimulate further advancements in this field of research.

## 2. Tumor Microenvironment and Tumor-Associated Macrophages (TAMs)

The immunological microenvironment in OS is complex, consisting of innate and adaptive immune cells, specifically macrophages and T lymphocytes, that play critical roles in tumor progression and response to therapy [8,12,13,14,15,16]. These immune cells can be found in the central and invasive edges of the tumor, as well as in the nearby tertiary lymphoid structures. A healthy immune system can identify malignant cells and launch functional and progressive processes to eradicate them. Patients with OS who have higher immunological scores and more immune cell infiltration in the microenvironment tend to have a more favorable prognosis [17,18]. Hence, it is crucial to prioritize the examination of the tumor immune microenvironment, particularly its interaction with tumor cells, to get a deeper comprehension of the OS immune system and facilitate the creation of innovative immunotherapies (Figure 2).

### Myeloid-Derived Suppressor Cells (MDSCs) and EMT

MDSCs are significant components in the TME of OS [10,20,21,22,23,24,25,26,27,28,29,30]. By secreting chemical substances such arginase-1 (Arg-1), inducible nitric oxide synthase (iNOS), and reactive oxygen species (ROS), MDSCs weaken anti-tumor immunity, especially T cell function, and make it easier for neoplastic cells to avoid immune surveillance [20,31,32,33,34]. In addition to their role in tumor angiogenesis, MDSCs secrete substances such as matrix metalloproteinase 9 (MMP9) and vascular endothelial growth factor (VEFG) [22,35,36,37,38]. In addition, MDSCs help immune evasion and tumor growth by interacting with other cells in the tumor microenvironment, like TAMs and Tregs [29]. In distant organs, MDSCs create pre-metastatic niches, which allow tumor cells to spread and colonize [30]. In addition, they release some growth factors (HGF and TGF-β), two substances that aid in the laborious process of EMT, which in turn promotes the invasion and dissemination of OS [31]. A new therapeutic target may be found in CD300ld, a surface molecule on MDSCs that has recently been found to play a key role in the immune-suppressive activity of MDSCs [39]. CD300ld is a protein specifically expressed on a subset of MDSCs, particularly the polymorphonuclear (PMN-MDSCs), which are a type of immune cell that plays a role in suppressing the immune response within a tumor microenvironment, making it a potential target for cancer immunotherapy; essentially, high CD300ld expression on MDSCs is associated with tumor progression and immune suppression [39]. Key points about CD300ld and MDSCs include the following: (1) CD300ld acts as a receptor on PMN-MDSCs, facilitating their recruitment to tumors and enhancing their ability to suppress T cell activity, promoting tumor growth; (2) Research indicates that high CD300ld expression on MDSCs is linked to poor prognosis in various cancers, making it a potential biomarker for tumor aggressiveness; (3) Targeting CD300ld through blocking antibodies could be a strategy to inhibit MDSC function and enhance anti-tumor immune responses, potentially improving the efficacy of cancer immunotherapy [39]. Thus, potential therapeutic targets include MDSCs due to their important roles in immune suppression and tumor development. Immune suppression and ineffective anti-tumor immune response could be alleviated with treatment strategies that target MDSCs. Nevertheless, further investigation is necessary prior to their use in clinical settings.

The immune component of the OS TME is mainly composed of tumor-associated macrophages (TAMΦs), which make up a substantial fraction relative to other immune cells [40]. TAMΦs are crucial in the inflammatory response and tissue homeostasis [41]. Macrophages exhibit a high degree of plasticity, adopting distinct phenotypes in response to environmental cues, with pro-inflammatory phenotype (M1) and the anti-inflammatory phenotype (M2) representing the two extremes of polarization [42]. M1 macrophages play a crucial role in suppressing tumors by activating the immune system to produce large amounts of pro-inflammatory cytokines such as IL-1, IL-6, and IL-12 [43]. They also contribute to the maturation of T helper type-1 (Th1) cells and the production of inducible nitric oxide synthase (iNOS) [19]. In contrast, M2 macrophages are linked to immune suppression, matrix degradation, and tumor angiogenesis, promoting tumor growth and metastasis. Multiple studies have demonstrated a strong correlation between a significant infiltration of TAMΦs and a poorer prognosis in most solid malignancies. Wolf-Dennen et al. discovered that in lung metastasis of OS, there was an elevated expression of M2-related cytokines, chemokines, and cell-markers [44]. In a similar vein, Dhupkar et al. demonstrated from an alternative standpoint that the transition from the M2 to M1 phenotype resulted in the regression of lung metastasis in OS, hence reinforcing the significance of M2 macrophages in the development of OS [45]. However, the role of TAMs in OS remains contentious, with conflicting reports regarding their impact on tumor progression and patient survival. While some research has demonstrated that a higher presence of TAMΦs is linked to decreased metastasis and improved survival rates in high-grade OS, conversely, the presence of M2 macrophages seems to indicate a poor prognosis [46,47]. It has been proposed that the balance between M1 and M2 macrophages, rather than the total quantity, may significantly impact OS prognosis [47].

Recent research has highlighted the potential of targeting TAMΦs as a therapeutic strategy regulating the course of OS. The aim is to control the polarization of TAMΦs to enhance the ratio of M1 phenotype to M2 phenotype or to induce the transition from M2 phenotype to M1 phenotype, thereby inhibiting the progression of OS. Immunomodulatory agents such as mifamurtide, which stimulate macrophages and enhance the production of proinflammatory cytokines [48], have received approval from the European Medical Agency for treating OS in conjunction with adjuvant chemotherapy [49,50,51]. Punzo et al. provided evidence of mifamurtide’s anti-tumor efficacy from two perspectives. In addition to altering macrophage polarization towards the intermediate M1-M2 phenotype, it can also limit the duration of M2 activation by reducing STAT3 and Akt phosphorylation, hence inhibiting the STAT3 pathway and PI3K/Akt/mTOR pathway, both of which promote M2 polarization [52]. Furthermore, scientific evidence has demonstrated that all-trans retinoic acid (ATRA) effectively inhibits the onset of OS and decreases the spread of OS to the lungs [53,54]. ATRA has the potential to indirectly alter the polarization of macrophages or disrupt the link between TAMs and cancer stem cells (CSCs) to restrict the production of CSCs in OS [53,54,55,56,57]. The anticipated effect of naturally occurring chemicals dihydroxy-coumarins (esculetin and fraxetin) was to promote a shift in TAM polarization from M2 to M1 phenotypes during OS therapy. Dihydroxycoumarins exert their effects by decreasing the synthesis and proliferation of IL-10, MCP-1, and TGF-β1. They also hinder the phosphorylation of STAT3, which activates M2 phenotypic differentiation [58]. Zoledronic acid is contemplated as a potential therapeutic medication due to its ability to disrupt M2 phenotype polarization and induce TAMΦs to revert to the M1 phenotype [59]. While the effectiveness of zoledronate treatment has been shown in certain bone metastasis cancers, randomized research indicated a greater risk than the placebo group [60]. Gomez-Brouchet et al. discovered that zoledronate could cause harmful polarization in CD68+/CD163+ bipotent macrophages [61]. CD68 is a widely recognized pan-macrophage marker in numerous tumor studies, despite being identified as a marker for M1-polarized macrophages. In contrast, a significant amount of CD163 staining is linked to elevated CMAF nuclear expression, a transcription factor for macrophages associated with M2 macrophage polarization [62]. Therefore, further investigation is required to determine the precise impact of zoledronate on OS treatment to ascertain if it is beneficial or detrimental.

Despite these advancements, the precise impact of TAMΦs modulation on OS treatment outcomes remains under investigation. Further research is needed to elucidate the mechanisms underlying TAMΦ-mediated immune regulation in OS and to determine the efficacy and safety of these therapeutic strategies.

## 3. Tumor-Infiltrating Lymphocytes (TILs)

Tumor-infiltrating lymphocytes (TILs) are immune cells that can be found in OS, and they may play a role in the treatment of this malignant bone cancer. TILs are the second most prevalent type of infiltrating immune cells in OS, present in about three-quarters (75%) of cases, with the highest occurrence of 86% observed in metastases [29,63,64,65,66,67,68,69,70,71,72,73,74,75,76,77]. These TILs consist of CD8+ T lymphocytes, CD4+ T lymphocytes, CD20+ B lymphocytes, and CD117+ mast cells [10]. Despite the modest fraction of CD8+ T cell infiltration in OS and that CD8+ T presence remains less common than myeloid cells in OS biopsies, suggesting low immunogenicity and tumor neo-antigen deficiency, there is a substantial positive association between their presence and the reduced rate of metastasis and improved survival outcome [26,29,68,78,79,80,81,82].

Ligon et al. identified immune resistance mechanisms within OS metastases, such as the upregulation of immune checkpoint proteins like TIM-3 and LAG-3 on TILs, which may impede their cytotoxic activity [29]. Data-driven mathematical modeling studies have shown a general increase in T cell populations in response to chemotherapy, with cytotoxic T cells, helper T cells, and dendritic cells mirroring the growth pattern of OS cells initially but decreased over time [83,84]. Conversely, regulatory T cells saw an initial decrease in the population, followed by an increase [85]. Fritzsching et al. demonstrated that OS patients with higher rates of CD8+/FOXP3+ ratio had significantly better survival rates [86,87,88]. Additionally, the ratio of CD8+/FOXP3+ T cells to regulatory CD4+/FOXP3+ T cells in biopsies taken before chemotherapy reveals patient survival.

Furthermore, immune-suppressive factors such as galectin-9 (Gal9), mainly expressed on CD4+CD25+ regulatory T cells (Tregs) in OS, contribute to immune evasion and tumor progression. The frequency of Gal9 expression on Tregs in OS was significantly higher than in non-cancer controls and higher than in other types of solid tumors [89]. Moreover, the Gal9 produced by CD4+CD25+ Tregs may have a role in the formation of M2 macrophages, resulting in a more potent inhibitory response from CD8+ T cells against tumors and the occurrence of inflammatory conditions [89]. T-cell depletion and induction of the exhausted state, characterized by the increased expression of immune checkpoints, further dampens the anti-tumor T-cell response [90]. Gao et al. observed a reduced ability of CD4+ T cells in individuals with OS to produce IL-21 compared to healthy controls. This decrease was particularly significant in follicular helper T (Tfh) cells, which typically exhibit high expression of PD-L1. This suggests a promising approach for immunotherapy in OS could involve reversing the decrease in IL-21 by reducing PD-L1 expression on Tfh cells or inhibiting the recruitment [40]. Other strategies that enhance T cell function, such as IL-21 supplementation to promote effector T cell growth and activity, or Treg-targeted therapies like RG6292 to deplete Tregs without disrupting IL-2 signaling in effector T cells, show promise in restoring anti-tumor immune responses [91,92].

Overall, understanding the intricate interplay between T lymphocytes and the OS microenvironment is essential for developing effective immunotherapeutic strategies to combat aggressive tumors.

## 4. Immune-Related Cells

Several non-immune cells can promote the development of tumors by regulating immune responses. Both normal tissue and tumor tissue-derived mesenchymal stem cells (MSCs) have been verified to promote OS progression [55]. The tumor-promoting action of MSCs can be ascribed to their ability to enhance the stemness qualities of OS cells through the action of IL-6, which promotes tumor growth [93]. Additionally, extracellular vesicles released by OS can shift MSCs towards a pro-tumor phenotype, characterized by a high production of IL-6 [94].

Another approach involves suppressing the immune response of activated MSCs, which includes inhibiting the growth of T cells, B cells, and NK cells, resulting in the development of Tregs [95]. OS is characterized by bone matrix remodeling, mainly facilitated by osteoclasts. These osteoclasts, which exhibit significant variability, originate from a monocytic lineage and can modulate T-cell activation like other immune cells derived from monocytic lineages, such as monocytes, macrophages, and dendritic cells. Their function is dictated by their progenitor cells and the surrounding milieu.

The involvement of osteoclasts in the development of OS is still a subject of debate. According to prior studies, a compelling concept proposes that osteoclasts facilitate the dissemination of tumor cells in the first phase but subsequently dismantle and rearrange bone environments. Osteoclasts play a vital role in regulating OS formation through their versatile immunological function. In neoadjuvant chemotherapy, the macrophage-phagocyte system engulfs drug-carrier particles and transports them to bone structures. Therefore, the osteoclasts, which are typically derived from macrophages in the bone microenvironment, near the OS can indicate the survival of macrophages and the amount of medication supplied to the bone marrow [96].

## 5. Immune Checkpoint Inhibitors (ICIs)

Within the TME, an increased expression of two immunological checkpoints, namely programmed cell death protein 1 (PD-1) [97,98] and cytotoxic T lymphocyte-associated protein 4 (CTLA-4) [99], was detected in T cells in most kinds of cancer (Table 1). Their activation enables immunological tolerance and resistance to therapy by blocking T-cell activation [100].

OS cells employ a deregulation strategy with a high ligand protein expression that activates specific pathways [101,102,103,104]. To prevent interaction between infiltrating T cells and tumor cells, ICIs are used to counteract immunological tolerance and stimulate an immune response against the tumor. Molecular mechanisms of PD-1 and CTLA-4 inhibition are among the earliest immune checkpoints investigated. These mechanisms include the induction of infiltration of CD8+ T-cells and ICOS+ Th1-like cells in the TME. In addition, using both PD-1 inhibition and radiation in combination therapy can significantly enhance the activation of CD8+ T cells and their migration to distant metastatic lesions, a phenomenon known as the abscopal effect.

CTLA-4 is a receptor made of a protein and attached to the cell membrane. It is found on Tregs and memory T cells. Its function inhibits the immune response to tumors by interacting with CD80/86 on dendritic cells. PD-1, a transmembrane immunoglobulin, is found on activated T cells and has a dual role: it inhibits CTLs and activates Tregs. Regarding PD-L1, its expression is identified on OS cells, indicating a decrease in the infiltration of immune cells such as T cells, NK cells, and dendritic cells, along with an increase in T-cell death [105]. Considering their promising outcomes in combating tumors, the FDA has approved the use of ICIs targeting CTLA-4 (ipilimumab), PD-1 (nivolumab, pembrolizumab, and cemiplimab), and PD-L1 (atezolizumab, avelumab, and durvalumab) in the treatment of various solid tumors. Initial generation ICIs that target CTLA-4, PD-1, and PD-L1 were employed in the treatment of OS [106].

These inhibitors can enhance the T-cell-mediated immune response against tumors. Nevertheless, clinical trials utilizing single-agent PD-1/PD-L1 ICIs did not reveal any predetermined outcomes. The SARC028 trial assessed the safety and effectiveness of pembrolizumab. Out of all the patients with skeletal sarcoma, only a tiny proportion (5%) had a measurable and positive response to treatment. A phase 2 clinical trial (NCT03013127) confirmed that pembrolizumab demonstrated acceptable tolerability but lacked significant clinical effectiveness in treating advanced OS in adult patients [107]. Additional evidence about the absence of ICI efficacy in OS was also indicated using nivolumab (NCT02304458) and atezolizumab (NCT02541604) in clinical trials, including juvenile patients. There was no favorable response in these trials, which included 23 patients with OS [96,108]. Furthermore, a clinical trial (NCT03359018) evaluating the combination of apatinib and camrelizumab did not provide any further advantages in terms of survival when compared to the use of apatinib alone, differently from the successful combination as seen in hepatocellular carcinoma, gestational trophoblastic neoplasia, and nasopharyngeal carcinoma. [109,110]. The limited efficacy in clinical trials with a single-agent ICI suggested that overall survival was predominantly unresponsive to PD-1/PD-L1 inhibition.

Combination therapy using immune cell activation enhancers in conjunction with ICIs has shown enhanced efficacy. Bempegaldesleukin, a novel agonist of the IL-2 pathway that specifically targets CD122 has been shown to significantly enhance the effectiveness of PD-1 and CTLA-4 ICIs in metastatic and orthotropic OS mouse models and seems to be promising in humans [111]. This enhancement is achieved by promoting the accumulation of effector T cells and Natural Killer cells inside the TME. Another factor contributing to the low objective response rate in PD-1/PD-L1 blocking therapy is the rapid depletion of tumor-targeted CTLs inside the TME, preventing them from exerting a sustained and potent anti-tumor impact [112]. Hence, it is evident that solitary PD-1/PD-L1 ICI therapy may not yield sufficient efficacy in treating OS. Studies using the K7M2 mouse model of metastatic OS treated with a combination of CTLA-4 and a PD-1 ICI showed effective control in most individuals, as seen by total tumor suppression [113]. Helm et al. discovered that in mice models with OS, a combination of CTLA-4 and PD-1 ICI increased CD8+ T cells. The Alliance A091401 trial assessed the efficacy of combining nivolumab with ipilimumab as a treatment for metastatic sarcoma. The results revealed that 16% of patients in the combination group responded positively to immunotherapy, whereas just 5% of patients in the nivolumab group had a verified reaction [114]. There is no conclusive evidence from clinical trials regarding the therapeutic impact of combining ICIs in OS. However, there have been some documented cases where the use of ipilimumab and nivolumab immunotherapy resulted in significant regression of tumor manifestations and stabilization of tumor mass in patients with metastatic OS [115,116].

However, challenges remain, including immune-related adverse events (irAEs) associated with ICIs, such as cytokine release syndrome. Combining cytokine receptor antagonists with ICIs may mitigate irAEs and improve prognosis. Furthermore, modulation of macrophage polarization in conjunction with PD-1/PD-L1 ICIs may overcome barriers to T-cell infiltration at metastatic sites.

## 6. CAR T and TCR T Therapies

Chimeric Antigen Receptor (CAR) T cells are genetically modified to identify and bind to antigens associated with tumors. In summary, the transduced CAR is composed of three primary components: (i) an ectodomain produced from a single chain variable fragment of an antibody that identifies explicitly a neoantigen, (ii) the transmembrane domain, and (iii) an endodomain containing intracellular signaling domains obtained from the CD3 ζ chain and co-stimulatory molecules. This arrangement allows T cells to identify tumor-associated antigens (TAAs) and induce the destruction of tumors by T-cell-mediated cytotoxicity, regardless of the major histocompatibility complex (MHC) [117]. Briefly, T cells are extracted from the patient’s blood. The T cells are modified in the laboratory to express a CAR protein (Genetic Engineering). The CAR protein is a combination of two parts: An antigen-binding domain that recognizes a specific protein on the surface of cancer cells and a signaling domain that activates the T cell to kill the cancer cell. The modified T cells are infused back into the patient (Infusion of CAR T Cells). The CAR T cells circulate in the body and identify cancer cells based on the antigen-binding domain. When they find cancer cells, they activate and release cytotoxic substances to kill them (Attack on Cancer Cells). CAR T cell therapy is currently approved by the FDA for the treatment of acute lymphoblastic leukemia (ALL), non-Hodgkin’s lymphoma (NHL), and multiple myeloma [118,119].

CD19-targeting CAR T cells have demonstrated an exceptional response against B cell hematologic malignancies, with up to 90% remission rates in clinical trials. As a result, they have been the first genetically modified cell-based treatment to receive FDA approval [120,121]. The limited targets identified due to tumor antigen variability in OS have limited the use of CAR T technology. Another reason for the failure to effectively treat solid tumors using CAR T cells is the limited ability of these cells to infiltrate and remain within the rigid osteoid bone tumor matrix, as well as the presence of immunosuppressive components in the tumor microenvironment [117]. Including costimulatory molecules and cytokines aims to enhance CAR T-cell activation and functionality. Clinical trials evaluating CD28-based and CD28-CD3ζ-OX40 CAR T cells specifically target patients with sarcoma (e.g., NCT00902044 and NCT01953900). Genetically modifying T cells to release cytokines selectively in response to tumor antigens can minimize systemic toxicity and increase cytokine concentration at the tumor site, enhancing therapeutic efficacy [122].

Unlike CAR T cells, T-cell Receptor (TCR) T cells express receptors derived from tumor antigen-specific T-cell clones with high affinity and acidity. This fact enables TCR T cells to exhibit selectivity and sensitivity in targeting human leukocyte antigens (HLA) on the surface of cells [123]. TCRs surpass antibodies and CARs in terms of targeting efficiency since they can infiltrate tumors and interact with intracellular and surface antigenic peptides provided by HLA [124,125]. This capability enables the identification of more hidden targets and expands the possibilities for future use in solid malignancies.

Cancer germline antigens are expressed in a limited manner in both testis tissues and tumor tissues with various histological origins. However, germ cells do not produce MHC molecules and are therefore shielded from immune attacks mediated by TCR T cells [126]. The cancer germline antigen NY-ESO-1 is a suitable TCR T cell therapy target. In a phase I/II trial evaluating NY-ESO-1-specific TCR-T therapy, 61% of patients with synovial cell sarcoma experienced therapeutic benefits without experiencing severe adverse effects [127]. Furthermore, NY-ESO-1 expression has been detected in 31.3% of OS tumors. There is ongoing testing of a TCR that explicitly targets NY-ESO-1 in patients with OS (NCT03462316). Prior investigations primarily utilized CD8+ T cells containing MHC-I-restricted TCRs for therapeutic purposes. However, Lu et al. successfully employed CD4+ T cells modified to express MHC-II-restricted TCRs and MAGE-A3 to treat patients with OS. Remarkably, all patients subjected to this experimental treatment exhibited objective partial responses concerning metastatic lung lesions [128]. Papillomavirus binding factor (PBF) is a DNA-binding transcription factor that is expressed in up to 92% of OS cases. The PBF A24.2 peptides have been found to stimulate CTLs in patients with HLA-A24-positive OS, leading to an immune response that targets and eradicates tumor cells [129]. A PBF TCR-multimer has been effectively generated to specifically recognize the naturally occurring PBF peptide on HLA-A24 + PBF + OS cells [130]. While there is currently no published evidence on utilizing PBF-modified T cells for treating OS, it is crucial to study this matter. This fact is supported by the promising outcomes observed with PBF’s impact on oncogenicity and immunogenicity.

However, the limits of TCR T treatment arise from the tumors’ ability to evade immune response by reducing the expression of their MHCs. Additionally, the utilization of this therapy is limited in patients with distinct HLA haplotypes due to HLA restriction. Another obstacle arises from the incorrect coupling of the newly added α/β TCR chains with the existing α/β TCR chains. This fact not only diminishes the production and effectiveness of TCR T cells but can also lead to the detection of unwanted antigens, potentially triggering an autoimmune response and causing organo-toxicities [131]. An ongoing clinical trial (NCT02508038) is now investigating the safety of transplanting TCRαβ+/CD19+-depleted haploidentical hematopoietic stem cells as a therapy for patients with OS.

Despite these strides, challenges persist, including limited T-cell infiltration, HLA restrictions, and tumor immune evasion mechanisms. Addressing these hurdles will be pivotal in unlocking the full potential of TCR T therapy in OS treatment.

## 7. Vaccine Approaches

Immunotherapy vaccines offer a promising avenue for OS treatment, with dendritic cell (DC)-based vaccines leading the forefront [132]. These vaccines harness the immune system’s power by leveraging dendritic cells to present tumor antigens to T cells, stimulating an antitumor response and eliminating immunosuppression [133,134]. The most recent development in DC vaccines is the FDA’s granting of orphan drug designation for ilixadencel in treating hepatocellular carcinoma. This classification was given because of the significant success observed in a Phase 1 trial conducted in Sweden using ilixadencel [135]. The typical manufacturing process for DCs-based vaccines involves the isolation of DCs from peripheral blood mononuclear cells, their maturation and exposure to tumor antigens outside the body, and their subsequent injection back into the patient [132]. DC-based vaccines are classified based on antigen pulsing methods, including peptides or proteins, DNA/RNA transfection, and tumor lysates [136]. Krishnadas et al. demonstrated a favorable clinical response using DCs loaded with peptides derived from cancer germline antigens (MAGE-A1, MAGE-A3, and NY-ESO-1) against OS and various other tumors such as neuroblastoma, Ewing sarcoma, and rhabdomyosarcoma [137].

In addition to DCs, two more types of immune cells, γδ T cells and macrophages, have been proposed for tumor vaccines. These vaccines are peptide-pulsed γδ T vaccines and chimeric antigen receptor macrophages (CAR-MΦs) [138]. Substantially, γδ T cells possess a high capacity to activate CD8+ T cells [139]. Furthermore, γδ T cells outperform conventional DC vaccinations because they can trigger cytotoxic activity against cancer cells in an HLA-independent manner [132]. In the preliminary investigation, the researchers observed that γδ T cells could directly identify and destroy OS cells, even though the specific cell lines targeted showed only a moderate level of vulnerability to γδ T cell cytotoxicity.

Table 2 presents the advantages and disadvantages of vaccine approaches against cancer.

Autologous tumor cell vaccines circumvent the need for isolating and culturing dendritic cells outside of the body, instead immediately triggering a dendritic cell response within the body. The process involves extracting tumor cells from the patient, expanding them if needed, subjecting them to irradiation, and reintroducing them into the patient [140]. Promising results were shown in patients with primitive neuroectodermal tumor (PNET) or Ewing sarcoma when treated with an autologous tumor cell vaccination, showing significant immune responses [141]. A recent investigation conducted a synergistic approach in oncology by combining an autologous tumor cell vaccine, ACT, with IL-2 in canines suffering from OS. The results demonstrated a significantly extended lifespan in comparison to the conventional treatment of amputation [142]. While these findings are promising, further research is needed to evaluate safety and efficacy in human patients with OS.

Novel methodologies are revitalizing tumor vaccination attempts. Peptide- and viral-based vaccinations share a common technique of directly delivering the antigen to the body’s DCs. PBF-derived peptide vaccine, which targets tumor-associated antigens, has been extensively studied in patients with OS who have the HLA-A2/A24+ genotype. Li et al. developed a peptide-specific tetramer targeting QVT and LSA peptides in a groundbreaking work. They demonstrated that this tetramer had lethal effects against HLA-A11 + PBF + OS cells [143]. Additional vaccinations utilizing HER2 have demonstrated advantages in canine models with OS. These include a HER2-targeted recombinant listeria vaccine and a peptide vaccine targeting the epidermal growth factor receptor (which shares homology with HER2). These vaccines have reduced metastasis and improved prognosis compared to the control group (189, 190). The peptide-based vaccination must operate within the framework of HLA-I, and an immunologic adjuvant is required to achieve an adequate T-cell response. In addition to its ability to induce polarization in macrophages, mifamurtide may also trigger TLR4 to increase the expression of type 1 interferon. This data suggests that it can be used as a replacement for INF-α [136]. Moreover, two recently concluded phase I/II clinical trials with oncolytic HSV1716 (NCT00931931) and unaltered oncolytic reovirus REOLYSIN^®^ (NCT00503295) investigated their efficacy in treating OS, potentially offering innovative strategies for oncolytic viral vaccines.

Table 3 presents the updated clinical trials from the official US governmental agency for registered clinical trials.

## 8. Conclusive Remarks

Advancements in our knowledge of the biological aspects of OS have led to the extensive development of treatment over the decades. Immunotherapy has significantly transformed the treatment of OS since its introduction, enabling the conversion of non-responders into responders and enhancing the existing responses. This research examined the distinct TME and present utilization of immunotherapy in OS, which may be categorized as ICIs, ACT, and cancer vaccines. Figure 3 presents a summary schema of the types of potential immunotherapies for OS.

Definitely, OS creates an immune-suppressive milieu with many M2 macrophages and few TILs. This environment promotes resistance to drugs and results in poor overall survival rates for OS patients. Immunotherapy specifically addresses the obstacles in tumor immune evasion and resistance to chemotherapy. ICIs focus on protein molecules or their ligands that suppress autoimmune activity, thereby reactivating the immune system. ACT generates exact and efficient tailored T cells in a laboratory setting. These lymphocytes are outfitted with receptors that specifically target neoplastic antigens. Tumor vaccines are biological substances that include specific TAA that aim to stimulate the immune system to identify and target these external antigens for attack.

While immunotherapy has demonstrated success in treating various cancers like melanoma, prostate cancer, renal cell carcinoma, and non-small-cell lung cancer, only a handful of innovative immunotherapies are currently available for OS. The primary challenges revolve around limited T-cell penetration and subsequent immunological toxicity. The infiltration of T-cells into targeted tissues is hindered by the immunosuppressive TME and thick fibrous tissue surrounding solid tumors.

Efforts to address these challenges include exploring the potential of oncolytic viruses, which infect and destroy tumor cells, to overcome resistance to PD-1 blocking therapy. Additionally, angiotensin inhibitors are being investigated to reduce the stiffening of the extracellular matrix in solid tumors, potentially improving drug delivery. Nanotechnology offers promising avenues, with biodegradable nanoparticles serving as adjuvants to regulate molecules at specific sites, enhance delivery, and stimulate T-cell growth outside the body.

The combination of various immunotherapeutic techniques, known as “immune cocktail therapy”, shows significant potential in influencing the cancer-immunity cycle, enhancing immune cell infiltration, and strengthening T-cell cytotoxicity. However, immunological agents can induce systemic cytokine release syndrome, capillary leak syndrome, sepsis-like syndrome, and immune-related adverse events (irAEs) that can affect any organ or tissue. Therefore, there is an urgent need for more individualized therapy strategies where patients can be categorized based on their predisposition to produce immune-related adverse events, using biomarkers to forecast patient reactions and monitor treatment progression. Individualized therapeutic strategies are urgently needed to minimize adverse effects, utilizing biomarkers to predict patient reactions and modify treatment agents based on biopsy samples and immune-related grading from clinical studies.

With regard to sarcomas harboring high inflammation, it is well-known that inflammation, which includes both the innate and adaptive immune systems, is the body’s way of protecting itself from potentially dangerous stimuli like damaged cells, irritants, infections, and even sterile lesions. This process helps to preserve tissue homeostasis. The inflammatory response that occurs during cancer development, in contrast to wound healing and infections, does not resolve. Additionally, it has been found that autoimmune diseases, bacterial and viral infections, obesity, smoking, asbestos exposure, and excessive alcohol consumption can all trigger tumor-extrinsic inflammation. These factors are known to raise the risk of cancer and speed up the progression of malignancies. However, tumor progression may occur as a result of cancer-initiating mutations that generate cancer-intrinsic or cancer-elicited inflammation. This inflammation is brought about by the recruitment and activation of inflammatory cells. It is well-known that immunosuppressive TME is a result of both extrinsic and intrinsic inflammation, making it an ideal environment for tumor formation. After the inflammatory TME has formed, inflammatory substances produced by tumor cells or interstitial cells activate oncogenes and then tumor suppressor genes, causing cells to proliferate and survive for a longer period of time. Washington University School of Medicine researchers in St. Louis conducted a clinical trial that demonstrated T cell immunotherapy, which involves genetically modifying patients’ own T cells to attack and kill cancer cells, effectively treats some patients with rare soft tissue cancers. Research published in The Lancet examined two uncommon malignancies: synovial sarcoma and myxoid round cell liposarcoma (MRCLS) [144].

Research has demonstrated that individuals with high-grade sarcoma who tested positive for either NY-ESO-1 or MAGE-A4 had a far superior overall survival rate compared to those who tested negative for these CTAs. The authors explained these findings by saying that many patients with synovial sarcoma and myxoid liposarcoma, which have generally excellent prognoses, were in the NY-ESO-1 and MAGE-A4 positive groups. The authors also provided an explanation for how and why the anti-tumor response was triggered by the expression of NY-ESO-1. However, it does not seem that everything can go smoothly and the TME of mild progressive STSs and highly aggressive sarcomas like UPS, MFS, and MPNST would have different roles for NY-ESO-1 and MAGEA4 according to another report [145]. Also, depending on the TME of naive T cells that encounter the antigen in highly aggressive STS T-cell dysfunction, such as functional memory or fatigue, tolerance, anergy, and senescence could happen. Various sarcomas should be studied in the future to determine the roles of NY-ESO-1 and MAGE-A4 in OS subtypes.

In conclusion, immunotherapy presents significant potential for treating OS, but further exploration of molecular pathways is essential to achieve more precise therapeutic outcomes.

## 9. Future Directions

The leading main bone malignancy in children and young adults is OS. While the prognosis for patients with localized OS has improved with current standard care, there has been no significant improvement in overall survival over the previous few decades. Even with aggressive treatment, the 5-year survival rate for metastatic or relapsed OS is only 20%. Consequently, improving the survival rate of individuals with OS requires immediate attention. Immunotherapy has the potential to be used to treat OS since the TME is immunosuppressive and contains cells that promote immunosuppression and express many immunosuppressive chemicals, particularly in the lung metastatic foci. Unfortunately, the lack of success with immunotherapy in OS is directly attributable to its distinct suppressive immune milieu. The therapy of OS with immune checkpoint blockers, such as anti-PD-1/PD-L1 and anti-CTLA-4 antibodies, has a modest therapeutic impact. Research on CAR-T treatment for OS remains in its infancy. Clinical trials for OS using CAR-T therapy have not yet yielded the same results as those for hematological malignancies, and research into additional CAR-engineered adoptive cell therapies, like CAR-NK and CAR-M, is still in its early stages. There is still a considerable amount of work to be done in applying immunotherapy to OS, according to the current preclinical and clinical data on the subject. In order to determine which subgroups would benefit most from immunotherapy and to forecast how well immunotherapy will work in patients, additional biomarkers must be investigated, the genetic background of the patients, and histological subtyping of the tumor may be critical at least for some variants or tumors harboring composite histological patterns. Despite OS having a low response rate to a single immunotherapy treatment, we strongly hope that one potential future direction is immunotherapy-based combination personalized therapy, which can transform a normal immune response into an inflammatory efficacious and effective microenvironment.

## Figures and Tables

**Figure 1 biomedicines-13-00664-f001:**
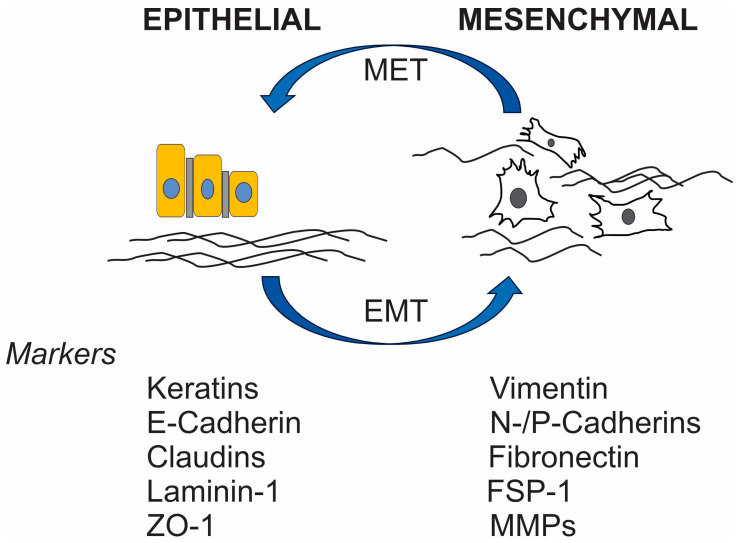
Epithelial and mesenchymal cell phenotypes with different differentiation markers. EMT, epithelial-mesenchymal transition; MET, mesenchymal-epithelial transition; FSP-1, fibroblast-specific protein 1; MMPs, matrix metalloproteinases; ZO-1, zonula occludens 1.

**Figure 2 biomedicines-13-00664-f002:**
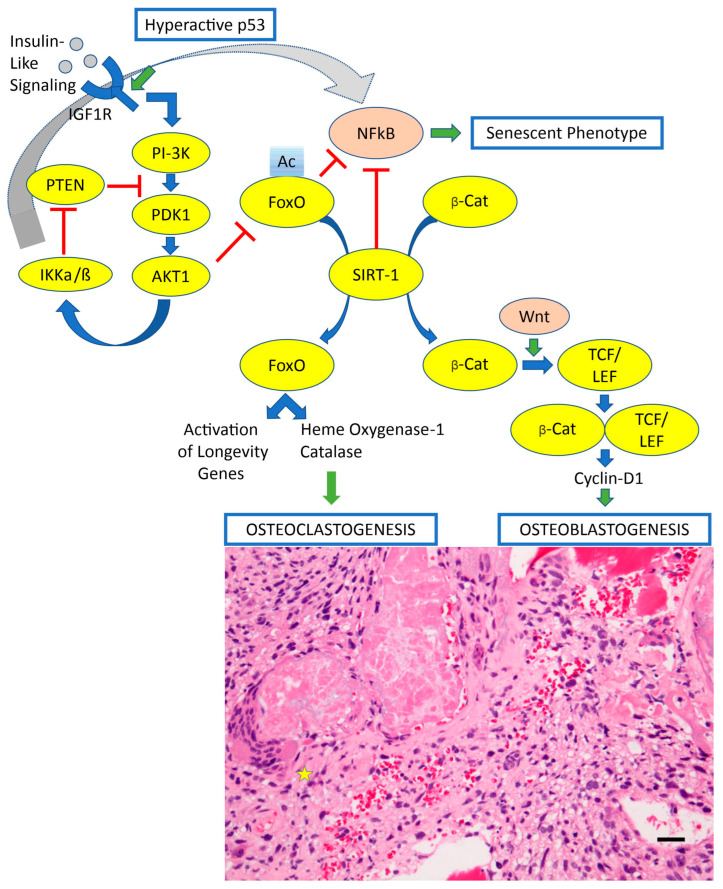
Signaling pathways contemplating insulin-like signaling/IGFR1, FOXOs, and SIRT1 for osteoclastogenesis and osteoblastogenesis. This picture also depicts the occurrence of a hyperactive p53. A hyperactive p53 would be a condition leading to OS-genesis. The abbreviations of this figure are shown in the text and please see the text for details. In both normal and abnormal states, the FOXO1, 3, and 4 proteins play an essential role in regulating bone mass and facilitating bone formation. In order to control osteoclast development and bone resorption, FOXOs play a crucial regulatory role by reducing ROS levels. Reduced trabecular and cortical bone mass results from increased proliferation, osteoclast production, and FOXO1–4, all of which are lost in osteoclast progenitors. In contrast, osteoclast differentiation is inhibited by FOXO3 gain-of-function. One example of an antioxidant enzyme that can inhibit H_2_O_2_ is catalase, whose increased expression leads to this feature. In general, HO-1 helps FOXOs prevent osteoclast formation. In addition, FOXOs hinder osteoblastogenesis by binding to and diverting β-catenin from TCF/LEF-mediated transcription, a process known as Wnt signaling. The result is a decline in bone mass after reductions in cyclin D1 and cell proliferation. On the other hand, it appears that FOXOs aid osteoblast survival by reducing oxidative cellular stress and raising the expression of the key antioxidant enzymes superoxide dismutase and catalase. The redox-active sulfhydryl moieties of the peptide glutathione help to lower ROS levels, while FOXO1 encourages their accumulation. The canonical route that controls FOXO transcriptional activity is the PI3K-PKB/AKT pathway. While FOXOs and SIRT1 theoretically contribute to bone lifespan by regulating the relative contributions of new bone formation and remodeling, the activity involving IGF1 and IGF-R1 may have the reverse effect. It seems that SIRT1 inhibits bone resorption and promotes bone formation by mediating posttranslational changes of FOXOs. Various kinases have the ability to phosphorylate FOXOs, and modifications made after translation may also impact FOXO activity. The microphotograph displaying the osteoblastogenesis and osteoclastogenesis in osteosarcoma originates from our original illustrative material published in an open source biomedical journal with Creative Common license distribution [19].

**Figure 3 biomedicines-13-00664-f003:**
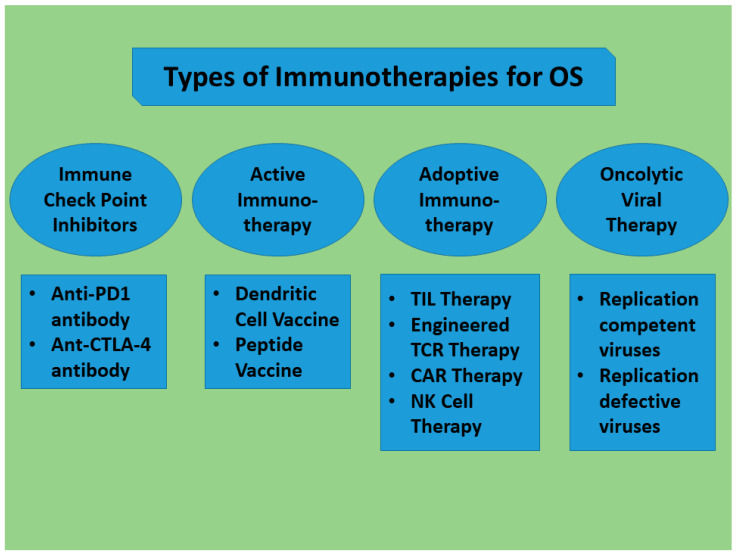
Types of potential Immunotherapies for OS. PD1, Programmed cell death protein 1; CTLA-4, Cytotoxic T-lymphocyte-associated protein 4; TIL, tumor-infiltrating lymphocyte; CAR, Chimeric Antigen Receptor T-cell Therapy; NK, natural killer.

**Table 1 biomedicines-13-00664-t001:** Immune Checkpoint Inhibitors (ICIs).

Class	Target	Drug	Indications
CTLA-4-I	CTLA-4	Ipilimumab	Melanoma
PD-1-I	PD-1	Cemiplimab	Metastatic cutaneous SqCC
		Nivolumab	NSCLC, SCLC, RCC, HCC, melanoma,
			Hodgkin’s lymphoma (CHL), HNC (SqCC),
			Metastatic CRC with high MSI or MMRD
			Urothelial carcinoma
		Pembrolizumab	NSCLC, LBCL (mediastinal), Hodgkin’s
			Lymphoma, gastric cancer, melanoma
			Cervical carcinoma, urothelial carcinoma
PD-L1-I	PD-L1	Avelumab	Merkel cell carcinoma, urothelial carcinoma
		Atezolizumab	NSCLC, urothelial carcinoma
		Durvalumab	NSCLC, urothelial carcinoma
CTLA-4-I + PD-1-I	CTLA-4 PD-1	Ipilimumab AND Nivolumab	CRC (some subtypes), RCC, melanoma

Notes: CRC, colorectal carcinoma; CTLA-4-I, cytotoxic T lymphocyte antigen 4 inhibitor; HCC, hepatocellular carcinoma; PD-1-I, programmed cell death 1 inhibitor; PD-L1-I, programmed cell death ligand 1 inhibitor; SqCC, squamous cell carcinoma; SCLC, small cell lung carcinoma; MMRD, mismatch-repair deficiency; MSI, microsatellite instability; NSCLC, non-small cell lung carcinoma; ICIs, immune checkpoint inhibitor; RCC, renal cell carcinoma.

**Table 2 biomedicines-13-00664-t002:** Classification of Vaccines.

Class	Advantages	Disadvantages
**DNA vaccine**	Simple and safe to manufacture and stable for storage, both cellular and humoral immunity, and long-term expression potential	Lack of positioning effect and targeting, gene integration with subsequent oncogenic potential, low uptake, transfection, and enzyme degradation
**RNA vaccine**	Safe, simple, and flexible to manufacture, no risk of genetic integration, and short half-life	Internal instability, low transfection efficiency
**Peptide vaccine**	Better safety and specificity, bypass the APC presenting process, simple, convenient, and economical preparation process	Weak immunogenicity for fewer epitopes, immunological tolerance for short peptides, delivery instability
**Tumor lysates vaccine**	Simple to manufacture without additional synthesis, it includes most of the potential tumor antigens	Immunosuppression inducer, weak antitumor immunity response due to interfering cellular content
**DC vaccine**	Distinct antitumor ability with better clinical efficacy	High costs with regard to treatment and manufacturing, low efficiency in vitro-induced DC mortality
**Tumor cell membrane vaccine**	Antitumor immunity promoter with multiple antigen motifs, avoids the interference of cell contents	Complexity with regard to the membrane structure limiting industrial and clinical applications
**ICD-induced antigen vaccine**	Strong operability, multiple induced methods, and ability to personalize vaccines (personalized vaccine therapy)	More energy consumption and cellular content interference

Notes: Table derived from text and illustrations from [122].

**Table 3 biomedicines-13-00664-t003:** US Governmental Registered Clinical Trials of Immunotherapy in Osteosarcoma (up to 18 February 2024).

NCT #	Name	Status	Res	Interventions	Phases	#	Type	Study Design	Start Date	End Date
06669013	A	RECR.	NO	Dinutuximab beta	3	40	INT.	Random., IM	5/20/2021	2025-09
06202599	B	COMPL.	NO	Fruquintinib	N.A.	124	OBS.	Observational Model	11/25/2021	11/15/2023
06171282	C	RECR.	NO	Oncolytic Recombinant HSV-1, R130	Early 1	9	INT.	SGIM	7/12/2023	7/12/2026
06114225	D	RECR.	NO	Metastasectomy and pre-op. IT (gemcitabine and penpulimab) and stereotactic body RT	2	43	INT.	IM	6/1/2023	12/30/2026
05851456	E	RECR.	NO	Oncolytic Rec.HSV-1, R130	Early 1	20	INT.	IM	4/10/2023	2026-04
05726383	F	RECR.	NO	DRUG: Iscador*P	2	32	INT.	IM	5/14/2024	5/11/2027
05660408	G	N.A.	NO	pp65 RNA	1–2	36	INT.	IM	2025-01	2035-10
05241132	H	N.A.	NO	Tislelizumab plus chemotherapy	2	27	INT.	IM	11/12/2021	10/31/2024
04751383	I	TERM.	NO	Dinutuximab, Magrolimab	1	12	INT.	IM	8/31/2021	9/30/2024
04730349	J	TERM.	YES	Nivolumab|, NKTR-214	1–2	15	INT.	NRIM	6/3/2021	6/22/2022
04616248	K	RECR.	NO	Anti-CD40 Agonist, Pembrolizumab, Tocilizumab	1	18	INT.	NRIM	1/9/2023	1/9/2027
04483778	L	N.A.	NO	Pembrolizumab	1	68	INT.	NRIM	7/13/2020	2040-12
04433221	M	N.A.	NO	Multiple sarcoma-specific CAR-T cells and sarcoma vaccines	1–2	20	INT.	IM	7/1/2020	12/31/2023
03842865	N	N.A.	NO	Vigil	N.A.	N.A.		N.A.		
03782363	O	N.A.	NO	Autologous CIK	1	0	INT.	NRIM	12/18/2020	4/1/2023
03635632	P	N.A.	NO	C7R-GD2.CART cells	1	94	INT.	NRIM	4/23/2019	5/16/2038
03618381	Q	RECR.	NO	2nd Gen 4-1BBÎ, EGFR806-EGFRt (CD19-Her2tG)	1	44	INT.	IM	6/18/2019	2040-06
03013127	R	TERM.	NO	DRUG: Pembrolizumab	2	12	INT.	IM	5/30/2017	1/31/2019
03006848	S	COMPL.	YES	Avelumab	2	19	INT.	IM	2/16/2017	3/18/2020
02982486	T	N.A.	NO	Ipilimumab, Nivolumab	2	60	INT.	IM	2017-12	2020-12
02173093	U	N.A.	NO	IL-2, GD2Bi-aATC, GM-CSF	1–2	40	INT.	IM	2014-11	2019-12
02107963	V	COMPL.	NO	Anti-GD2-CAR engineered T cells, AP1903, Cyclophosphamide	1	15	INT.	NRIM	2/28/2014	1/31/2017
02100891	W	COMPL.	NO	Allogeneic HCT (Donor NK Cell Infusion)	2	15	INT.	IM	3/20/2013	7/15/2020
00001564	X	COMPL.	NO	EF-1 Peptide, EF-2 Peptide, PXFK Peptide, E7 Peptide, IL-2, IL-4, GM-CSF, CD40 Ligand	2	30	INT.	IM	12/23/1996	10/25/2007
00001566	Y	COMPL.	YES	Autologous dendritic cells (indinavir sulfate)	2	42	INT.	IM	1996-12	2008-09

**Notes**: #, number; The titles of the 25 clinical trials follow the first 25 letters of the 26 letters of the English Alphabet (A B C D E F G H I J K L M N O P Q R S T U V W X Y Z) and are as follows: (A) Chemo-immunotherapy in Patients Under 18 Years of Age With Bone and Soft Tissue Sarcomas; (B) Fruquintinib-based Treatment for Refractory Bone and Soft Tissue Sarcomas After Several Lines of TKIs’ Resistance; (C) A Clinical Study on Oncolytic Virus Injection (R130) for the Treatment of Advanced Bone and Soft Tissue Tumors; (D) Pulmonary Resectable Osteosarcoma Treated by Metastasectomy and Pre-operative Immunotherapy and Stereotactic Body Radiotherapy (PROMIS): a Prospective Clinical Trial; (E) A Clinical Study on Oncolytic Virus Injection (R130) for the Treatment of Relapsed/Refractory Bone and Soft Tissue Tumors; (F) IscadorÂ^®^ P (Mistletoe) Immunotherapy for Recurrent Osteogenic Sarcoma; (G) RNA Lipid Particles Targeting Pediatric Recurrent Intracranial Malignancies and Other systemic Solid Tumors; (H) Tislelizumab Combined With Chemotherapy in the Treatment of Bone Metastases of Unknown Primary; (I) Testing the Combination of Two Immunotherapy Drugs (Magrolimab and Dinutuximab) in Patients With Relapsed or Refractory Neuroblastoma or Relapsed Osteosarcoma; (J) A Study of Bempegaldesleukin (BEMPEG: NKTR-214) in Combination With Nivolumab in Children, Adolescents, and Young Adults With Recurrent or Treatment-resistant Cancer; (K) In Situ Immunomodulation with CDX-301, Radiation Therapy, CDX-1140, and Poly-ICLC in Patients W/Unresectable and Metastatic Solid Tumors; (L) B7H3 CAR T Cell Immunotherapy for Recurrent/Refractory Solid Tumors in Children and Young Adults; (M) Combination Immunotherapy Targeting Sarcomas; (N) Expanded Access of Vigil in Solid Tumors; (O) Study of Adoptive Immunotherapy in Relapsed and Non-resectable Sarcomas After Multimodal Treatment; (P) C7R-GD2.CART Cells for Patients With Relapsed or Refractory Neuroblastoma and Other GD2 Positive Cancers (GAIL-N); (Q) EGFR806 CAR T Cell Immunotherapy for Recurrent/Refractory Solid Tumors in Children and Young Adults; (R) A Study of Pembrolizumab in Patients With Relapsed Or Metastatic Osteosarcoma Not Eligible for Curative Surgery; (S) A Phase II Trial of Avelumab in Patients With Recurrent or Progressive Osteosarcoma; (T) A Phase II of Nivolumab Plus Ipilimumab in Non-resectable Sarcoma and Endometrial Carcinoma; (U) Activated T Cells Armed With GD2 Bispecific Antibody in Children and Young Adults With Neuroblastoma and Osteosarcoma; (V) A Phase I Trial of T Cells Expressing an Anti-GD2 Chimeric Antigen Receptor in Children and Young Adults With GD2+ Solid Tumors; (W) Phase 2 STIR Trial: Haploidentical Transplant and Donor Natural Killer Cells for Solid Tumors; (X) A Pilot Study of Tumor-Specific Peptide Vaccination and IL-2 With or Without Autologous T Cell Transplantation in Recurrent Pediatric Sarcomas; (Y) A Pilot Study of Autologous T-Cell Transplantation With Vaccine Driven Expansion of Anti-Tumor Effectors After Cytoreductive Therapy in Metastatic Pediatric Sarcomas; N.A., not available, withdrawn, not yet recruiting, or unknown; INT, interventional; OBS, observational; IM, Intervention Model; SGIM, single group intervention model; NRIM, non-randomized intervention model; Recr., recruiting; Res., results available.

## Data Availability

All data are available online and the Clinical Register of the Clinical Trials is available online. If any documentation is needed or cooperation is required, please contact the corresponding author.

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
