# Peer review of "Digging Through the Complexities of Immunological Approaches in Emerging Osteosarcoma Therapeutics: A Comprehensive Narrative Review with Updated Clinical Trials"

_biomedicines, 2025, doi:10.3390/biomedicines13030664_

Round 1

Reviewer 1 Report (Previous Reviewer 2)

Comments and Suggestions for Authors

The Authors aimed to describe immunotherapies for osteosarcoma.

The topic is interesting.

The narrative nature of this review must be acknowledged. Fig 1 seems not useful and confusing.

Is there any study about these therapies in Osteosarcoma? This is of paramount importance and should be reported in a related table.

Author Response

The Authors aimed to describe immunotherapies for osteosarcoma.

The topic is interesting.

The narrative nature of this review must be acknowledged. Fig 1 seems not useful and confusing.

Is there any study about these therapies in Osteosarcoma? This is of paramount importance and should be reported in a related table.

Thank you very much for your comments and suggestions. The manuscript was thoroughly revised, considering all the comments and suggestions of all reviewers. In particular, the title was changed to highlight the narrative review of the manuscript. Figure 1 was fully edited. We deleted the features part because we agree with you that talking about anoikis would be confusing. The epithelial cells were detached from each other, and a better presentation of the adherence of the epithelial cells is shown.

Yes, we went to the US Government Agency for registered clinical trials and updated all clinical trials dealing with osteosarcoma until February 18, 2025.

Thank you again for the opportunity to revise the manuscript. We will submit a clean version and a version highlighting the changes.

Reviewer 2 Report (New Reviewer)

Comments and Suggestions for Authors

The manuscript entitled Emerging Therapeutics and Immunologic Approaches for Skeletal Osteogenic Sarcoma. A Comprehensive Review , aims to explore the potential of immunotherapy for osteosarcoma through an examination of its immune microenvironment. However, there are also some problems that need to be addressed.

1.This review predominantly focuses on the immune microenvironment of osteosarcoma and various immunotherapeutic approaches, as highlighted in the title, abstract, and throughout the majority of the manuscript. Nonetheless, the discussion on tumor cell migration and epithelial-mesenchymal transition (EMT) is limited to a minor section of the paper. It is recommended to consider the necessity and relevance of including this section.

2. The abstract places significant emphasis on the treatment of osteosarcoma through immunotherapeutic approaches; however, it frequently references the mechanisms of tumor dissemination, which undermines the coherence of this section.

3. Lines 259-261: CD68 is widely recognized as a pan-macrophage marker in numerous tumor studies. Reference 65, as cited by the author, elaborates on the diverse roles of CD68 as a macrophage marker. However, this manuscript exclusively presents CD68 as a marker of M1 polarization, which may lead to potential misinterpretations.

4. The manuscript's description of significant literature findings, particularly in lines 398-400, lacks appropriate source citations. It is advisable to meticulously review other sections of the manuscript for similar omissions.

5. Line 794: This section delineates various categories of vaccines. Subheadings may be established based on the distinct content to enhance the clarity of the presentation.

6. Line 795: The abbreviation for "osteosarcoma" has been previously defined in the preceding section and has been utilized in its abbreviated form in subsequent sections. Therefore, reiterating this definition is unnecessary.

Author Response

The manuscript entitled “Emerging Therapeutics and Immunologic Approaches for Skeletal Osteogenic Sarcoma. A Comprehensive Review” , aims to explore the potential of immunotherapy for osteosarcoma through an examination of its immune microenvironment. However, there are also some problems that need to be addressed.

1.This review predominantly focuses on the immune microenvironment of osteosarcoma and various immunotherapeutic approaches, as highlighted in the title, abstract, and throughout the majority of the manuscript. Nonetheless, the discussion on tumor cell migration and epithelial-mesenchymal transition (EMT) is limited to a minor section of the paper. It is recommended to consider the necessity and relevance of including this section.

  1. The abstract places significant emphasis on the treatment of osteosarcoma through immunotherapeutic approaches; however, it frequently references the mechanisms of tumor dissemination, which undermines the coherence of this section.
  2. Lines 259-261: CD68 is widely recognized as a pan-macrophage marker in numerous tumor studies. Reference 65, as cited by the author, elaborates on the diverse roles of CD68 as a macrophage marker. However, this manuscript exclusively presents CD68 as a marker of M1 polarization, which may lead to potential misinterpretations.
  3. The manuscript's description of significant literature findings, particularly in lines 398-400, lacks appropriate source citations. It is advisable to meticulously review other sections of the manuscript for similar omissions.
  4. Line 794: This section delineates various categories of vaccines. Subheadings may be established based on the distinct content to enhance the clarity of the presentation.
  5. Line 795: The abbreviation for "osteosarcoma" has been previously defined in the preceding section and has been utilized in its abbreviated form in subsequent sections. Therefore, reiterating this definition is unnecessary.

Thank you very much for your comments and suggestions. The manuscript was thoroughly revised, considering all the comments and suggestions of all reviewers. 

We tried to shorten the manuscript, but considering the emphasis required by other reviewers on TME, we expanded some sections and balanced the manuscript according to your suggestions.

Yes, thank you. The abstract was completely rewritten and revised, considering your suggestions and the suggestions gathered from other reviewers.

Yes, thank you. We expanded the sentence dealing with CD68.

The citations in the manuscript were updated, and more references were added according to your suggestions and those of other reviewers.

The vaccine section was expanded, and a table with advantages and disadvantages was added.

In all parts of the manuscript, the abbreviation OS replaced the occasional occurrence of the word "osteosarcoma."

Thank you again for the opportunity to revise the manuscript according to your suggestions and those of other reviewers.

Reviewer 3 Report (New Reviewer)

Comments and Suggestions for Authors

This review explores the tumor microenvironment of OS and the potential of immunotherapy. In particular, it highlights the important role of immune cells, including tumor-associated macrophages and T cells, in OS progression and therapeutic response. Immunotherapy may provide more effective and less adverse outcomes by preventing drug resistance and reducing immune escape. While very interesting, there are some concerns.

Please elaborate a bit more on the latest clinical trials.

Please review how immunotherapy works in sarcomas with high inflammation.

If you would like, I can refer you to the following literature.

Clinicopathological Assessment of Cancer/Testis Antigens NY-ESO-1 and MAGE-A4 in Highly Aggressive Soft Tissue Sarcomas. Diagnostics (Basel). 2022 Mar 17;12(3):733. doi: 10.3390/diagnostics12030733. PMID: 35328286; PMCID: PMC8946957.

Please state your FUTURE DIRECTION In Immunotherapy for sarcoma.

Please share any new findings of this review that you have seen and how it differs from other reviews.

Author Response

This review explores the tumor microenvironment of OS and the potential of immunotherapy. In particular, it highlights the important role of immune cells, including tumor-associated macrophages and T cells, in OS progression and therapeutic response. Immunotherapy may provide more effective and less adverse outcomes by preventing drug resistance and reducing immune escape. While very interesting, there are some concerns.

Please elaborate a bit more on the latest clinical trials.

Please review how immunotherapy works in sarcomas with high inflammation.

If you would like, I can refer you to the following literature.

Clinicopathological Assessment of Cancer/Testis Antigens NY-ESO-1 and MAGE-A4 in Highly Aggressive Soft Tissue Sarcomas. Diagnostics (Basel). 2022 Mar 17;12(3):733. doi: 10.3390/diagnostics12030733. PMID: 35328286; PMCID: PMC8946957.

Please state your “FUTURE DIRECTION In Immunotherapy for sarcoma”.

Please share any new findings of this review that you have seen and how it differs from other reviews.

Thank you very much for your comments and suggestions. The manuscript was thoroughly revised, considering all the comments and suggestions of all reviewers.

In particular, we went to the register website for clinical trials registered with the U.S. Government and added a table summarizing the current clinical trials and the terminated ones until February 18, 2025.

Moreover, we expanded the paragraph about inflammation with the article published in Lancet and the reference you suggested (

Clinicopathological Assessment of Cancer/Testis Antigens NY-ESO-1 and MAGE-A4 in Highly Aggressive Soft Tissue Sarcomas. Diagnostics (Basel). 2022 Mar 17;12(3):733. doi: 10.3390/diagnostics12030733. PMID: 35328286; PMCID: PMC8946957.

We also added the section "Future Directions," as suggested.

Many thanks for the opportunity to revise the manuscript according to your suggestions and other reviewers' suggestions.

Round 2

Reviewer 3 Report (New Reviewer)

Comments and Suggestions for Authors

The authors replied well, so the manuscript is suitable for publication.

Comments on the Quality of English Language

Good.

This manuscript is a resubmission of an earlier submission. The following is a list of the peer review reports and author responses from that submission.

Round 1

Reviewer 1 Report

Comments and Suggestions for Authors

The review by Sergi et al has aimed to discuss immunologic approaches for Osteosarcoma.

General comments.

1. This review is extremely difficult to read, multiple sections are written as large slabs of text without paragraphs or a defined structure. This has resulted in a review that is not concise or clear in its message. 

2. The authors have put too much emphasis on immunology background instead of relating results to osteosarcoma (OS). 

3. Multiple sections have little or no context to osteosarcoma and should be removed eg vaccines for bony cancer, artificial intelligence. 

4. Authors should remove 50% of the text and only discuss approaches/results related to osteosarcoma. Use tables as well to summarise findings.

Minor comments

1. Line 26. What is meant by "genuine" heterogeneity 

2. Line 27: The comment regarding funding should be removed. All cancer agencies should be interested. 

3. Line 47: Expand on the "remarkable improvements"

4. Immunotherapy has worked exceptional well for "hot" high mutation burden tumours. No discussion of this is apparent.  Driver genes such as p53, Rb1, CDKN2A and PTEN, and aberrant signaling in the PI3K/mTOR, should be discussed (PMID: 37894336, Tange et al., 2023)

5. Lines 61-66: Comment is not specific to Osteosarcoma, and is an inherent problem for all adolescent/pediatric cancers. This section can be condensed as no discussion of the mental, economically issues in relation to immunotherapy are discussed.

6. Lines 77-79: References are required

7. Line 83: Reference required

8. Immune checkpoint inhibitors. Table does not show any clinical trials for OS hence should be removed. 

9. Lines 364-375: Remove

10:  Authors should present in a table current/past immunotherapy clinical trials involving OS patients and outcomes

Comments on the Quality of English Language

It is clear that multiple people have written different sections of this review. Sentence structure, clarity and tone for the abstract and introduction does not match the rest of the manuscript. 

Reviewer 2 Report

Comments and Suggestions for Authors

The Authors aimed to describe novel therapies and immunotherapies for osteosarcoma.

The topic is interesting.

The narrative nature of this review must be acknowledged. Also, the title should be modified into "osteosarcoma" and please detail about "emerging therapies".

Fig 2 legend not clear. Please detail further.

Would resume the paragraphs describing the tumor microenvironment. Also, a table might be helpful.

Paragraph 7 does not completely relate to the topic.

In conclusions, although the topic is interesting, the paper is disorganized and too long in all its parts. It is difficult to follow as too many details are reported even in repetitions. Please have it shortened and put as much information in tables.

Is there any study about these therapies in Osteosarcoma? This is of paramount importance and should be reported in a related table.